# Multi-Agent Reinforcement Learning with Communication-Constrained Priors

**Guang Yang**[1]    **Tianpei Yang**[12*]    **Jingwen Qiao**[2]    **Yanqing Wu**[2]    **Jing Huo**[1]
**Xingguo Chen**[4]    **Yang Gao**[123*]
[1]State Key Laboratory for Novel Software Technology, Nanjing University
[2]School of Intelligence Science and Technology, Nanjing University
Accepted [3]School of Network Security and Information Technology, YiLi Normal University
[4]Nanjing University of Posts and Telecommunications
{yangg,jingwenqiao,yanqingwu}@smail.nju.edu.cn
{tianpei.yang,huojing,gaoy}@nju.edu.cn
chenxg@njupt.edu.cn

## Abstract

Communication is one of the effective means to improve the learning of cooperative policy in multi-agent systems. However, in most real-world scenarios, lossy communication is a prevalent issue. Existing multi-agent reinforcement learning with communication, due to their limited scalability and robustness, struggles to apply to complex and dynamic real-world environments. To address these challenges, we propose a generalized communication-constrained model to uniformly characterize communication conditions across different scenarios. Based on this, we utilize it as a learning prior to distinguish between lossy and lossless messages for specific scenarios. Additionally, we decouple the impact of lossy and lossless messages on distributed decision-making, drawing on a dual mutual information estimatior, and introduce a communication-constrained multi-agent reinforcement learning framework, quantifying the impact of communication messages into the global reward. Finally, we validate the effectiveness of our approach across several communication-constrained benchmarks.

## 1 Introduction

In multi-agent reinforcement learning (MARL) with partial observations, collaboration poses a significant challenge [19]. Communication is one of the effective measures to improve the learning of cooperative policy in MARL [31], widely applied in scenarios such as autonomous driving [4, 24] and cooperative drones [3, 12]. However, real-world scenarios are far from ideal and communication between agents often faces various constraints, specifically: (1) limited communication bandwidth, meaning only a limited amount of message can be transmitted, and (2) lossy communication, where transmitted message may be subject to interference, delay, loss, and other issues.

Much of the research on communication-constrained multi-agent reinforcement learning (MARL) focuses on the limited bandwidth issue [22, 14, 10, 30]. In this setting, it is assumed that the communication channel is ideal, which means that transmission is real-time and lossless. Therefore, these methods typically only need to focus on how to effectively allocate communication resources (e.g., communication bandwidth, communication medium) to promote cooperation among agents. For example, compressing the communication information to extract and transmit parts beneficial to cooperation [16], and systematically allocating communication mediums to avoid competition, among other problems [11].

---

*Corresponding Author: Tianpei Yang; Yang Gao

However, in most real-world scenarios, communication links are uncertain, and lossy communication is more prevalent. In this setting, existing work typically addresses two types of issue: noise interference and communication delay. Approaches to addressing noise interference typically involve modeling the unknown noise distribution and constructing learnable processes to adaptively optimize cooperative policy [7]. Another more common issue is communication delay, which frequently occurs in wireless network environments, which refers to the non-real-time transmission of message, It has been shown to impact the performance of multi-agent behaviors [29, 27]. Approaches to addressing communication delay focus primarily on how to remove the impact of delayed message on cooperative policy, such as constructing communication buffer [29] and determining when to communicate [27]. Nevertheless, the above methods are all based on ideal assumptions about communication delay, and may not be applicable to more complex and unknown real-world scenarios such as underwater and caves. The reasons are as follows: (1) These methods *lack scalability due to the lack of consideration of the common characteristics of lossy communication in different unknown scenarios*. (2) These methods *lack robustness due to the lack of consideration of the dilemma of promoting the relevance of effective communication messages and suppressing the relevance of lossy communication information*.

In order to overcome the above challenges, we first propose a generalized model of lossy communication to uniformly characterize the communication conditions in different scenarios, such as underwater, caves, and wireless networks. Based on this model, for specific scenarios, we use the lossy communication model as a learning prior to differentiate between lossy and lossless messages. Furthermore, for the second issue, we decouple the impact of different types of messages on distributed decision-making, borrowed from dual mutual information estimatior. On one hand, we enhance the positive impact of lossless messages on decision-making by maximizing the lower bound of mutual information. On the other hand, we reduce the negative impact of lossy messages on decision-making by minimizing the upper bound of mutual information. Finally, we propose a communication-constrained MARL framework and validate its robust performance in two communication-constrained scenarios serving as benchmarks: Markov Model-Based and Distance-Based communication constraints.

## 2    Related Work

The challenge of multi-agent collaboration within communication-constrained environments has undergone extensive scholarly investigation. Existing research predominantly addresses two key dimensions: one concerns the optimization of interactions in bandwidth-constrained scenarios, and the other focuses on enhancing system robustness under conditions of message loss.

### 2.1    MARL with Communication Bandwidth Constraints

Research on communication bandwidth constraints has primarily focused on how to achieve efficient interaction among multiple agents under limited bandwidth resources. Zhang et al. [28] proposed the VBC method, which reduces communication overhead by constraining the variance of information exchanged between agents during the training phase, thereby eliminating noise from the messages. Kim et al. [11] addressed the issue of medium contention in the information transmission process, proposing the SchedNet framework. By incorporating a MAC protocol from the wireless communication field, this framework alleviates bandwidth pressure and addresses the scheduling problem of constrained agents. Mao et al. [18] introduced the Gated-ACML algorithm, which uses a gating mechanism to filter messages. It adaptively prunes unnecessary information by determining whether a message is beneficial using a threshold, thus reducing the number of messages exchanged. Hu et al. [9] proposed the ETCNet framework, which converts limited bandwidth into a penalty threshold for event-triggered strategies, improving communication efficiency in multi-agent systems by sending messages only when necessary.

These methods essentially perform effective compression on messages, which aligns closely with the information bottleneck [21] principle—eliminating task-irrelevant redundant information and extracting task-relevant refined information. Such compression approaches have been validated to exhibit a certain degree of generalization capability in similar tasks [25]. However, compression is not equivalent to robustness [6, 23]. In environments with unstable or lossy communication, these compressed methods may not be effective. For instance, unstable communication is highly prone to losing critical messages, leading to a significant degradation in the quality of policies.

## 2.2 MARL with Lossy Communication

Research on lossy communication has primarily focused on the robustness of multi-agent systems under non-ideal communication conditions, such as environmental noise, transmission delays, and packet loss. Freed et al. [7] proposed a novel differentiable communication method using a randomized message encoding scheme, where it mathematically equates discrete communication channels to simulated channels with additive noise, enabling gradient backpropagation through these channels. Kim et al. [13] proposed the Message-Dropout training method, based on the concept of Dropout, which randomly drops communication messages from other agents during training to improve robustness against communication errors during execution. Zhang et al. [29] introduced the TMC method, which leverages temporal locality to reduce redundant messages and incorporates a message buffering mechanism to enhance robustness against packet loss, making it suitable for bandwidth-constrained and packet-loss-prone network environments. Yuan et al. [27] proposed DACOM, an adaptive delay-aware multi-agent communication model in which agents can learn to schedule waiting times for messages from other agents, which is suitable for delay-sensitive tasks and high-latency scenarios.

Nevertheless, the aforementioned studies predominantly remain confined to communication-constrained problems in specific scenarios. While these research approaches help to analyze particular issues in depth, its conclusions may not fully apply to other more complex and dynamic real-world application scenarios. To better align with real-world scenarios, in the next section, we have further refined the formalization of MARL with communication constraints.

## 3 Problem Formulation

Considering a fully collaborative multi-agent task where each agent is in a partially observable and communicative environment, it can be modeled as a decentralized partially observable Markov decision process (Dec-POMDP) [1] with communication, which is defined by the following 9-tuple, $\mathcal{G} = \langle \mathcal{N}, \mathcal{S}, \mathcal{O}, \mathcal{A}, \mathcal{T}, \mathcal{R}, \mathcal{Z}, \mathcal{M}, \mathbb{I}, \gamma \rangle$. $\mathcal{N} = \{1, \ldots, N\}$ is denoted by the set of $N$ agents. $\mathcal{S}$ is denoted by the set of environmental states. $\mathcal{O} = \{o^i\}_{i=1}^N$ is observation set of all agents, and $o^i$ is an observation set for agent $i$, which is determined by observation function $\mathcal{Z}(s, i)$. $\mathcal{A} = \mathcal{A}^1 \times \cdots \times \mathcal{A}^N$ is the set of agents' joint action space, where $\mathcal{A}^i$ is denoted by the action space of agent $i$. $r = \mathcal{R}(s, a)$ is the global reward signal shared by the agents. $\mathcal{T} : \mathcal{S} \times \mathcal{A} \times \mathcal{S} \to [0, 1]$ is denoted by the transition probability. $\gamma \in [0, 1)$ is the discount factor.

Furthermore, $\mathcal{M}$ is denoted by the message space, and $m^{ij} \in \mathcal{M}$ is denoted by the message sent by agent $i$ to agent $j$. To uniformly characterize the communication conditions in different environments, we further introduce a notation $\mathbb{I} = \{\iota^{ij}\}_{i \neq j}$, where $\iota^{ij} \in \{0, 1\}$ is denoted by the communication link status when agent $i$ sends message to agent $j$, where 1 indicates effective communication, while 0 indicates lossy communication. The set of message received by the agent $i$ can be defined by $M^i = \{\iota^{ji} m^{ji}\}_{j=1}^{j \neq i}$. In this case where the communication link is dynamic, the information received by each agent is different.

Then, given observation $o_t^i$ and message $M^i$ at the time step $t$, each agent $i$ uses a stochastic policy $\pi^i(\cdot | o^i, M^i)$ to choose actions. We denote the joint policy as $\pi = \{\pi^1, \pi^2, \cdots, \pi^N\} \in \Pi$, where $\Pi$ is the joint policy space. In cooperative MARL, the collaborative team aims to find a joint policy to maximize the total expected discounted return $J(\pi) = \mathbb{E}_\pi \left[ \sum_{t=0}^\infty \gamma^t r_t \right]$.

## 4 Robust Learning with Communication-Constrained Priors

To overcome two challenges of multi-agent collaborative policy learning with lossy communication, this section will introduce our algorithmic framework, which specifically includes: (1) modeling communication-constrained priors to capture the dynamics of communication links; (2) estimating messages' behavioral impacts to characterize the correlation between different communication messages and agent behaviors; and (3) proposing a communication-constrained MARL approach to optimize and enhance the robustness of policy learning across diverse communication environments.

## 4.1 Communication-Constrained Priors Modeling

In communication-constrained MARL, modeling constrainted communication in unknown scenarios is crucial and challenging. It is necessary not only to abstract the common problems that affect multi-agent policy learning as much as possible, but also to generalize them in different real-world scenarios. Therefore, we first propose a binary communication link parameter $\iota$ to characterize message reliability. In addition, for different scenarios, the communication link can be further formalized as follows,

$$\iota^{ij} = f_{\theta_e}(s^{ij}), \tag{1}$$

where $\theta_e$ is the parameter determined by the environment, and $s^{ij}$ is the part of the state most relevant to agents $i$ and $j$. For different environments, $f_{\theta_e}$ can be defined manually or obtained through pre-training of binary classification tasks. In the context of specific learning, there are several approaches: (1) When addressing specific and stable communication-constrained environments, one can estimate the priors of communication links through sampling or empirical data, enabling policy learning to adapt as closely as possible to these specific conditions. (2) In the face of diverse and non-stable communication-constrained environments, designing more diverse priors for communication links can allow policy learning to cover multiple exceptional scenarios. Specifically, message-dropout [13] is considered a case of this prior modeling, where messages are randomly masked with a certain probability to adapt to communication constraints. Therefore, incorporating such priors helps distinguish more effectively between lossy and lossless messages.

## 4.2 Messages' Behavioral Impacts Estimating

Building on the previous section, a natural goal is to maximize the utilization of lossless messages while minimizing the adverse effects of lossy messages. It is critical to measure the correlation between messages and agent behaviors, where we will specifically characterize via mutual information.

### 4.2.1 MI between Messages and Behaviors

MI is often used to improve multi-agent collaboration in MARL [22, 15]. It is a theory that measures the correlation between different variables. When extended to communication-constrained MARL, a basic requirement is to measure the correlation between messages and agent behaviors. So we have

$$I(m^{ji}, a^i) = H(a^i) - H(a^i|m^{ji}), j \neq i \tag{2}$$

where $H(\cdot)$ and $H(\cdot|\cdot)$ denote the entropy and conditional entropy respectively, and $m^{ji}$ denote the message transmitted from agent $j$ to agent $i$. $H(a^i)$ describes the ability to explore various behaviors of agent $i$, which could help generate diverse trajectories and avoid policy collapse when maximized. $H(a^i|m^{ji})$ measures the behavioral uncertainty of agent $i$, which encourages agent $i$ to behave deterministically given message $m^{ji}$ when minimized.

### 4.2.2 Du-MIE for Constrained Communication

Note that the basic definitions of communication-constrained prior and MI, we thus formulate a natural objective propose for communication-constrained MARL: maximize the MI between lossless messages and agent behaviors (to enhance their positive correlations) while minimizing the MI between lossy messages and agent behaviors (to mitigate their negative correlations). However, exact computation of MI is highly challenging, as it necessitates simultaneously determining both the joint probability distribution and the marginal distributions of the variables involved [2]. Inspired by the Dual Mutual Information Estimatior (Du-MIE) [15], we construct a Du-MIE for constrained communication to estimate the above objective. The Jensen-Shannon MI estimator based on Jensen-Shannon divergence (JSD) [8] estimates the lower bound of $I(m^{ji}; a^i)$ for maximization and the Contrastive Log-ratio Upper Bound (CLUB) [5] estimates the upper bound of $I(m^{ji}; a^i)$ for minimization, based on replay buffer $D$ with lossless messages $\mathcal{M}^+$ and lossy messages $\mathcal{M}^-$ respectively. Specifically, JSD can be defined as

$$I_{\text{JSD}}(m^{ji}; a^i) = \underbrace{\mathbb{E}_{\mathbb{P}_{\mathcal{M}\mathcal{A}^i}}\left[-\text{sp}\left(-T_{\theta_1}(m_t^{ji}, a_t^i)\right)\right] - \mathbb{E}_{\mathbb{P}_{\mathcal{M}^i}\otimes\mathbb{P}_{\mathcal{A}^i}}\left[\text{sp}\left(T_{\theta_1}(m_t^{ji}, a_k^i)\right)\right]}_{-\mathcal{L}^{ji}(\theta_1)}, \tag{3}$$

where $\mathbb{P}_{\mathcal{M}\mathcal{A}^i}$ is message-action joint distribution for agent $i$, $\mathbb{P}_{\mathcal{M}^i} \otimes \mathbb{P}_{\mathcal{A}^i}$ is the product of the marginals. $\text{sp}(z) = \log(1 + e^z)$ is the softplus function. $T_{\theta_1}$ is a discriminator function modeled by a

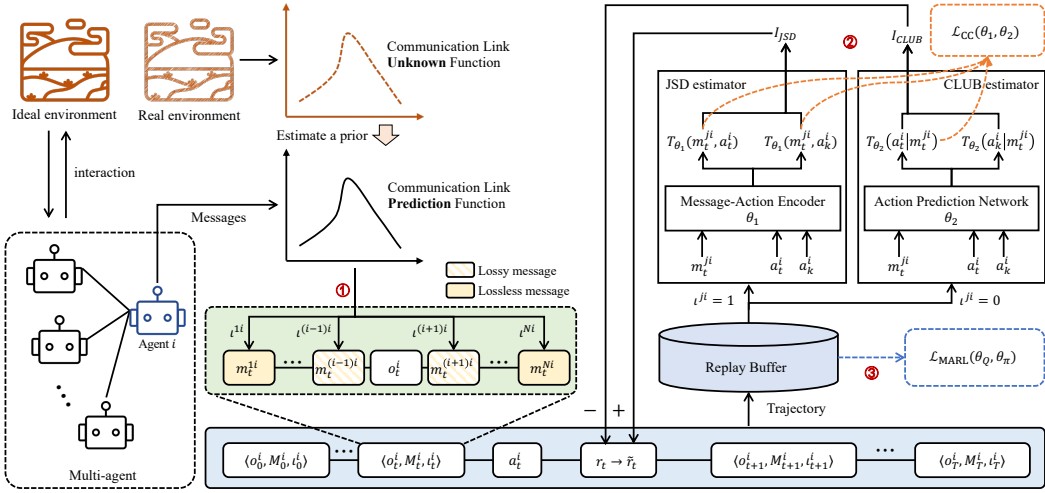

Figure 1: The overall framework for communication-constrained MARL. It can be divided into three main steps: ① Distinguishing between lossy and lossless messages by constructing communication link priors; ② Shaping the global reward through learning Du-MIE for constrained communication; ③ Stably optimizing multi-agent policies based on MARL algorithms.

neural network with parameters $\theta_1 \in \Theta$. $m_t^{ji}$ and $a_t^i$ are obtained by joint sampling from the replay buffer $\mathcal{D}$ with lossless messages $\mathcal{M}^+$, while $a_k^i$ is sampled independently from the same buffer. By minimizing the loss $\mathcal{L}^{ji}(\theta_1)$, we can make $I_{\text{JSD}}(m^{ji}; a^i)$ closely approximate the lower bound of $I(m^{ji}; a^i)$ [8]. On the contrary, CLUB can be defined as

$$I_{\text{CLUB}}(m^{ji}; a^i) = \underbrace{\mathbb{E}_{\mathbb{P}_{\mathcal{M}\mathcal{A}^i}} \left[ \log T_{\theta_2}(a_t^i | m_t^{ji}) \right]}_{-\mathcal{L}^{ji}(\theta_2)} - \mathbb{E}_{\mathbb{P}_{\mathcal{M}^i} \otimes \mathbb{P}_{\mathcal{A}^i}} \left[ \log T_{\theta_2}(a_k^i | m_t^{ji}) \right], \tag{4}$$

where $T_{\theta_2}$ is a variational approximation modeled by a neural network with parameters $\theta_2 \in \Theta$. The samples are sampled in the same way as JSD, but are based on the replay buffer $\mathcal{D}$ with lossy messages $\mathcal{M}^-$. Similarly, by minimizing the loss $\mathcal{L}^{ji}(\theta_2)$, we can make $I_{\text{CLUB}}(m^{ji}; a^i)$ closely approximate the upper bound of $I(m^{ji}; a^i)$.

Note that the training losses of the JSD and CLUB estimators are $\mathcal{L}^{ji}(\theta_1)$ and $\mathcal{L}^{ji}(\theta_2)$ respectively, then, the overall training loss of Du-MIE for constrained communication is defined as

$$\mathcal{L}_{\text{CC}}(\theta_1, \theta_2) = \sum_i \sum_{j \neq i} \iota^{ji} \mathcal{L}^{ji}(\theta_1) + (1 - \iota^{ji}) \mathcal{L}^{ji}(\theta_2). \tag{5}$$

The architecture of Du-MIE for constrained communication are illustrated in 1. After training, the JSD and CLUB can be used as learning signals to guide the behavior of the agent towards lossless communication and away from lossy communication, which will be introduced in the next subsection.

### 4.3 Communication-Constrained MARL

In order to characterize the impact of the upper and lower bounds of mutual information on multi-agent policy learning, inspired by reward shaping [15], we reshape the global reward as follows:

$$\tilde{r}_t = r_t + \sum_i \sum_{j \neq i} \alpha \iota^{ji} I_{\text{JSD}}(m^{ji}; a^i) - \beta(1 - \iota^{ji}) I_{\text{CLUB}}(m^{ji}; a^i) \tag{6}$$

where $\alpha, \beta$ are weight coefficients. Then, a new goal in the communication-constrained MALR is to maximize the expected discounted return $\tilde{J}(\pi) = \mathbb{E}_\pi \left[ \sum_{t=0}^\infty \gamma^t \tilde{r}_t \right]$.

This reward shaping can be combined with different MARL algorithms. Taking CTDE-based MARL algorithms as an example, the results of the combination are presented in Algorithm 1. Both policy-based [17] and value-based [20] methods can derive a global temporal difference (TD) loss, which is

presented as follows:

$$\mathcal{L}_{\text{MARL}}(\theta_Q) = \mathbb{E}_{s_t, a_t, r_t, s_{t+1} \sim \mathcal{D}} \left[ \left( Q_{\theta_Q}(s_t, a_t) - \left( \tilde{r}_t + \gamma Q_{\theta_Q^-}(s_{t+1}, a_{t+1}) \right) \right)^2 \right], \tag{7}$$

where $Q_{\theta_Q}$ is the parameters of the global Q-value network and $Q_{\theta_Q^-}$ is the parameters of the corresponding target network. For value-based MARL, the joint actions $a_{t+1}$ are obtained via greedy policies $a_{t+1} = \arg\max_{a_{t+1}} Q_{\theta_Q^-}(s_{t+1}, a_{t+1})$. For policy-based MARL, the joint actions $a_{t+1} = \{a_{t+1}^i\}_{i=1}^N$ are obtained via target policies $a_{t+1}^i \sim \pi_{\theta_{\pi^i}^-}(\cdot|o_{t+1}^i, M_{t+1}^i)$, and an additional policy loss (in actor network) must be incorporated, as follows:

$$\mathcal{L}_{\text{MARL}}(\theta_\pi) = \mathbb{E}_{s_t, a_t \sim \mathcal{D}} \left[ -Q_{\theta_Q}(s_t, a_t) \right]. \tag{8}$$

Lines 5–13 present the sampling process of trajectories. Compared to traditional MARL, the samples here not only include additional communication messages and the communication links but also reshape the global reward. Line 14 shows the Du-MIE training according to the equation (5). Finally, in line 15, to adapt to different MARL algorithms, the loss function should be correspondingly adjusted, as described in equations (7) and (8).

---

**Algorithm 1** Communication-Constrained MARL
___
1: **Input:** maximum episode length $T$, hyperparameters $\alpha$ and $\beta$ to balance the effects of MI, update frequency $k$ for Du-MIE, communication-constrained priors $f_{\theta_e}$.
2: **Initialize:** main network parameters in MARL $\theta_Q, \theta_\pi$, corresponding target networks $\theta_Q^-$ and $\theta_\pi^-$, JSD parameters $\theta_1$, CLUB parameters $\theta_2$.
3: **Initialize:** experience replay buffer $\mathcal{D}$.
4: **repeat**
5:     **for** $t = 1$ **to** $T$ **do**
6:         Get patial observation $o_t = \{o_t^i\}_{i=1}^N$.
7:         Get message $M_t = \{M_t^i\}_{i=1}^N$
8:         Predict communication link status $\mathbb{I}_t = \{\iota_t^i\}_{i=1}^N$, $\iota_t^i = \{\iota^{ji}\}_{j \neq i}$.
9:         Execute joint actions $a_t = \{a_t^i\}_{i=1}^N$ via sampling $a_t^i \sim \pi^i(\cdot|o_t^i, M_t^i)$.
10:        Receive $o_{t+1} = \{o_{t+1}^i\}_{i=1}^N$, $M_{t+1} = \{M_{t+1}^i\}_{i=1}^N$ and team reward $r_t$.
11:        Calculate the shaping reward $\tilde{r}_t$, according to the equation (6).
12:     **end for**
13:     Store $v = \{o_t, M_t, \mathbb{I}_t, o_{t+1}, M_{t+1}, a_t, \tilde{r}_t\}_{t=1}^T$ to $\mathcal{D}$.
14:     Update Du-MIE with replay buffer $\mathcal{D}$ every $k$ steps, according to the equation (5).
15:     Update network parameters in MARL, $\theta_Q, \theta_\pi$, according to the equations (7) and (8).
16: **until** reaching maximum training steps
___

## 5 Experiments

In this section, we evaluate the algorithm's effectiveness from three aspects: overall performance, the impact of communication-constrained priors (CCPs), and the role of Du-MIE for messages. The specific evaluation contents are as follows:

**(1) Overall Performance:** This aims to validate the proposed algorithm's performance under different communication-constrained scenarios.

**(2) Impact of Communication Priors:** This focuses on verifying the performance and properties of the proposed method when using different communication priors.

**(3) Role of Du-MIE for Messages:** Through ablation experiments, this evaluation seeks to determine how this module impacts the learning of multi-agent policies under communication constraints.

### 5.1 Experimental Setup

In the experimental setup, we integrate the proposed algorithm framework with MADDPG [17] to form Communication-Constrained MADDPG (CC-MADDPG) as the primary validation target. It is

then compared with four baselines: MAIC [26], Full-Communication MADDPG (FC-MADDPG), Dropout-MADDPG, and the standard MADDPG, operating without inter-agent communication. We adopt the Multi-Agent Particle Environments (MPEs) [17] as benchmarks. To simulate communication constraints, we employ the following two distinct models,

**Markov-Based Communication (MBC):** The Markov model [29] assumes that the state of a system at any time is determined by a state transition probability matrix, which includes one noiseless state and multiple lossy states—the more the number of lossy states, the higher the loss probability. In this experiment, the transition probability matrices are set with dimensions of $3, 6$ and $8$, corresponding to loss levels of light, medium, and heavy, respectively.

**Distance-Based Communication (DBC):** This approach [27] simulates signal attenuation in real-world environments (e.g., underwater or cave-like) based on inter-agent distances. Specifically, it sets a distance threshold to determine the degree of communication constraint, where smaller distances lead to higher message loss rates. In this experiment, the distance threshold are set with $5, 3$ and $1$, corresponding to loss levels of light, medium and heavy, respectively.

## 5.2 Results and Analysis

### 5.2.1 Performance Evaluation

This section systematically evaluates the overall performance and robustness of the CC-MADDPG algorithm by analyzing average episode cumulative rewards across different MPE task scenarios and communication-constrained testing environments.

Table 1 compares the performance (mean and standard deviation) of our algorithm with baseline methods across various task scenarios and communication constraints.Experimental results demonstrate that while MADDPG with communication achieves satisfactory performance in ideal communication environments, it exhibits high sensitivity to communication quality. Any form of communication constraint leads to significant performance degradation. MADDPG shows the lowest performance across all testing environments, with its average rewards substantially inferior to other communication-based algorithms, emphasizing the critical role of effective inter-agent communication for efficient collaboration in MPE tasks. For the Dropout-MADDPG algorithm, performance varies with different packet loss rates: the dropout-0.2 configuration generally achieves relatively better results, while higher dropout rates (e.g., 0.8) sometimes cause performance deterioration, as evidenced by its mere 36 average reward in Simple_Reference scenarios and consistent underperformance compared to FC-MADDPG in most environments, suggesting that excessive message dropout during training undermines agents' ability to learn effective collaborative policies.

In contrast, CC-MADDPG consistently achieves average rewards comparable to or exceeding other algorithms across both ideal and constrained communication environments. Notably, it maintains superior performance even under extreme communication conditions like heavy distance-based constraint (approaching non-communication scenarios), exemplified by its 138.0 performance in Simple_Tag when FC-MADDPG deteriorates to 1.5, demonstrating remarkable robustness. Compared with multiple communication-based baselines, MAIC attains the lowest average performance across all four tasks and is largely insensitive to the strength of communication constraints, indicating that the incentive messages it produces fail to effectively guide cooperation in these tasks.

An interesting observation emerges from our time-varying packet loss environments: performance variations across different network fluctuation levels (light, medium, and heavy) remain relatively insignificant. It may stem from two factors: (1) The predefined Markov-based packet loss patterns, while simulating bursty and correlated wireless channel characteristics, might generate insufficient average loss rates or inadequate durations of "bad states" to persistently disrupt critical collaboration moments, potentially mitigated by inherent robustness in agent policies; (2) The fixed 25-step episode length in MPE tasks may limit full manifestation of time-varying channel impacts.

### 5.2.2 Impact of Communication Constraint Priors

This section investigates the effectiveness of incorporating communication constraint priors during the multi-agent reinforcement learning training phase, and analyzes how different prior strategies specifically influence model performance in communication-constrained environments.

Table 1: Performance Comparison of Multi-Agent Algorithms Under Communication Constraints

| Task Scenario | Testing Environment | Algorithms | | | | | | |
|---|---|---|---|---|---|---|---|---|
| | | MAIC | FC-MADDPG | Dropout-MADDPG | | | MADDPG | CC-MADDPG |
| | | | 0.2 | 0.5 | 0.8 | | | |
| Simple_Tag | Unrestricted | 1.5±4.9 | 75.9±65.3 | 70.3±65.9 | 65.9±62.0 | 72.1±66.1 | 5.2±12.8 | **134.7±89.9** |
| | Light MBC (3) | 1.8±6.0 | 72.9±65.3 | 70.9±65.9 | 65.9±62.1 | 72.5±66.1 | 5.2±12.8 | **133.6±89.9** |
| | Medium MBC (6) | 1.6±8.6 | 67.2±53.7 | 71.0±65.7 | 67.1±62.3 | 72.1±66.1 | 5.5±15.6 | **134.9±90.9** |
| | Heavy MBC (8) | 1.7±6.2 | 54.4±43.8 | 69.8±66.5 | 67.9±63.2 | 72.7±66.0 | 6.0±14.2 | **131.4±86.2** |
| | Light DBC (5) | 2.3±7.8 | 19.5±38.2 | 69.0±65.8 | 67.3±62.4 | 70.8±64.2 | 23.5±41.5 | **136.9±89.7** |
| | Medium DBC (3) | 2.4±6.5 | 10.9±26.5 | 70.1±65.8 | 67.8±62.4 | 70.7±63.8 | 44.8±58.4 | **135.3±85.1** |
| | Heavy DBC (1) | 2.7±9.1 | 1.5±5.2 | 68.7±64.3 | 66.3±60.5 | 71.4±65.7 | 71.2±70.9 | **138.0±88.1** |
| Simple_Spread | Unrestricted | -298.0±75.1 | -138.7±26.0 | -145.4±23.5 | -137.5±27.6 | -138.1±22.8 | -194.9±26.7 | **-129.4±20.1** |
| | Light MBC (3) | -293.3±63.8 | -138.8±25.5 | -145.1±23.2 | -138.1±27.6 | -136.1±22.7 | -193.5±26.7 | **-129.2±19.9** |
| | Medium MBC (6) | -295.9±78.4 | -138.7±25.4 | -145.4±22.5 | -138.0±27.6 | -136.0±22.5 | -192.9±27.1 | **-129.0±20.2** |
| | Heavy MBC (8) | -283.7±51.8 | -142.0±26.9 | -144.3±24.2 | -138.1±27.4 | -135.8±22.5 | -190.9±27.7 | **-128.7±19.2** |
| | Light DBC (5) | -301.2±65.5 | -138.7±24.4 | -138.8±23.4 | -145.6±27.6 | -138.1±22.0 | -190.5±24.7 | **-128.7±20.3** |
| | Medium DBC (3) | -282.4±63.8 | -156.3±24.9 | -139.7±21.2 | -143.4±27.0 | -138.0±22.9 | -177.4±25.6 | **-127.6±19.2** |
| | Heavy DBC (1) | -289.5±63.1 | -191.9±26.8 | -140.9±20.8 | -144.5±27.6 | -138.0±23.5 | -169.6±27.1 | **-128.0±20.6** |
| Simple_Reference | Unrestricted | 0.2±1.4 | 51.0±62.6 | 54.4±78.3 | 51.6±69.0 | 38.0±57.9 | 2.4±4.1 | **76.9±76.8** |
| | Light MBC (3) | 0.3±1.7 | 47.2±62.6 | 54.3±78.3 | 51.6±66.4 | 37.3±58.5 | 2.4±4.1 | **76.5±76.8** |
| | Medium MBC (6) | 0.2±1.4 | 42.3±53.2 | 54.4±78.0 | 51.8±67.9 | 38.5±58.3 | 2.7±4.0 | **75.5±75.8** |
| | Heavy MBC (8) | 0.2±1.3 | 27.9±35.5 | 54.3±78.2 | 54.9±67.3 | 36.3±58.3 | 2.7±4.3 | **76.2±77.9** |
| | Light DBC (5) | 0.4±2.0 | 49.3±62.3 | 53.3±77.4 | 53.0±70.3 | 36.0±58.3 | 23.5±4.1 | **75.5±76.6** |
| | Medium DBC (3) | 0.3±1.4 | 39.1±56.5 | 53.3±76.4 | 53.3±71.9 | 37.2±59.4 | 32.0±4.2 | **73.8±74.5** |
| | Heavy DBC (1) | 0.1±1.0 | 4.0±12.9 | 52.8±76.5 | 53.2±70.7 | 34.6±60.3 | 41.0±4.1 | **62.1±70.1** |
| Simple_Adversary | Unrestricted | -29.5±26.4 | -6.7±5.1 | -6.8±5.0 | -6.6±4.8 | -6.7±4.8 | -75.5±48.5 | **-5.8±5.0** |
| | Light MBC (3) | -33.9±26.7 | -6.7±5.1 | -6.8±5.0 | -6.6±4.8 | -6.7±4.8 | -75.5±48.5 | **-5.9±5.0** |
| | Medium MBC (6) | -35.4±29.5 | -7.1±5.0 | -7.2±5.1 | -6.6±4.9 | -6.7±4.8 | -62.9±41.6 | **-6.1±5.0** |
| | Heavy MBC (8) | -33.1±28.7 | -7.5±5.0 | -6.8±5.0 | -6.8±4.9 | -6.7±4.7 | -51.5±34.5 | **-6.1±5.1** |
| | Light DBC (5) | -34.9±31.0 | -26.6±15.1 | -10.8±13.3 | -6.3±5.0 | -6.7±4.7 | -14.0±8.0 | **-6.4±5.8** |
| | Medium DBC (3) | -29.3±25.4 | -38.6±21.3 | -22.4±19.0 | -6.9±5.2 | -6.7±4.7 | -8.8±5.8 | **-7.7±7.2** |
| | Heavy DBC (1) | -35.0±29.3 | -45.3±21.8 | -27.3±19.5 | -7.7±5.3 | -6.8±4.7 | -7.8±5.5 | **-8.8±7.4** |

Table 2: Performance Comparison of CC-MADDPG with Different Priors

| Task Scenario | Testing Environment | Priors Type | |
|---|---|---|---|
| | | dropout-0.2 | Test-Matched |
| Simple_Spread | Light DBC | -128.7±20.3 | **-119.0±29.4** |
| | Medium DBC | -127.6±19.2 | **-98.0±30.5** |
| | Heavy DBC | -128.0±20.6 | **-107.0±33.3** |
| Simple_Reference | Light DBC | 75.5±76.6 | **84.2±84.1** |
| | Medium DBC | 73.8±74.5 | **107.5±109.1** |
| | Heavy DBC | 62.1±70.1 | **80.2±86.7** |

As evidenced by comparative results in Table 1, algorithms introducing communication constraint priors during training demonstrate significant advantages. Using FC-MADDPG as a baseline reference, although it achieves high performance under ideal testing conditions, its performance suffers catastrophic degradation when encountering any form of communication constraints. For instance, in the Simple_Tag scenario under distance-based constraints, FC-MADDPG's average episode cumulative reward plummets from 75.9 in ideal conditions to 19.5, 10.9, and 1.5 respectively. This drastic performance deterioration highlights the vulnerability of models trained under ideal conditions when facing communication constraints.

In stark contrast, algorithms incorporating communication constraint priors during training, such as Dropout-MADDPG and CC-MADDPG, exhibit superior robustness. Specifically, Dropout-MADDPG with dropout-0.2 prior achieves an average reward of 68.7 in the heavy distance-based constraint environment of Simple_Tag, vastly outperforming FC-MADDPG's 1.5. Remarkably, CC-MADDPG utilizing the same dropout-0.2 prior even surpasses its ideal communication environment performance with a score of 138. These results conclusively demonstrate that introducing communication constraint priors during training enables better adaptation to constrained communication environments during testing, yielding enhanced performance stability.

To ensure satisfactory generalization in unknown and variable testing environments, CC-MADDPG in this study adopts dropout-0.2 as the standard communication constraint prior by default. This

generalized random message dropout prior covers multiple potential communication constraint patterns, endowing the model with fundamental robustness. Building on this foundation, we further investigate whether employing training priors that precisely match the actual test environment constraints could yield additional performance improvements. Verification experiments reveal in Table 2 that models trained with "test-environment-matched" priors (where training constraints exactly match testing constraints) consistently outperform those using generic dropout-0.2 priors. For example, in Simple_Spread's medium distance-based constraint environment, the generic prior model achieves -127.6 average reward versus -98 for the matched prior model. These findings indicate that while generic message dropout priors provide baseline robustness, precisely tailored priors reflecting target deployment environments can substantially optimize model performance.

### 5.2.3    Impact of Dual Mutual Information Optimization Module

This section investigates the effectiveness of the proposed dual mutual information optimization module through ablation studies. To ensure fairness in comparisons, all algorithm variants in this ablation study employ identical communication constraint priors (dropout-0.2). We designed four algorithm variants for systematic comparison:

**Baseline Model**: Excludes all mutual information optimizations, equivalent to the MADDPG with dropout-0.2 communication constraint prior during training.

**Variant 1**: Activates only the lossless messages utilization component in Du-MIE module, which employs a JSD-based MIE to maximize the lower bound of mutual information between agent decisions and valid messages.

**Variant 2**: Activates only the lossy messages suppression component in the Du-MIE module, which only utilizes a CLUB-based MIE to minimize the upper bound of mutual information between agent decisions and invalid messages.

**Full Model**: Represents our complete proposed algorithm with both optimization directions activated in Du-MIE for messages.

Table 3 summarizes the average episode cumulative rewards of these four variants across various communication-constrained testing environments in Simple_Tag and Simple_Spread scenarios. Between the baseline and single-component variants, both mutual information optimizations independently enhance performance. In Simple_Tag, the baseline achieves 70 average rewards across environments. Variant 1 (maximization-only) elevates rewards to 81, while Variant 2 (minimization-only) significantly boosts performance to 120. This confirms the intrinsic value of mutual information optimization beyond prior knowledge utilization.

The full model consistently outperforms all variants, demonstrating synergistic complementarity between the two optimization components. For instance, in Simple_Tag's most constrained short-distance environment, the full model achieves 138.0 rewards versus 68.7 (baseline), 81.7 (Variant 1), and 120.4 (Variant 2). Similar patterns emerge in Simple_Spread, where the full model (-127.6 to -128.0) substantially surpasses the baseline (-143.4 to -145.6). These results validate that dual-directional mutual information optimization synergistically enhances multi-agent collaboration and system robustness under communication constraints.

Table 3: Ablation Study on Dual Mutual Information Module

| Scenario | MI Coefficients | | Average Episode Cumulative Reward | | | | | | |
|---|---|---|---|---|---|---|---|---|---|
| | min | max | Unrestricted | Light MBC | Medium MBC | Heavy MBC | Light DBC | Medium DBC | Heavy DBC |
| Simple_Tag | 0 | 0 | 70.3±65.9 | 70.9±65.9 | 71.0±65.7 | 69.8±66.5 | 69.0±65.8 | 70.1±65.8 | 68.7±64.3 |
| | 0 | 0.01 | 81.3±75.2 | 81.3±75.2 | 81.7±77.0 | 81.6±74.0 | 81.8±74.5 | 81.7±75.6 | 81.7±73.3 |
| | 0.001 | 0 | 120.5±82.1 | 119.8±81.0 | 120.3±81.8 | 119.3±78.3 | 121.5±84.5 | 119.5±80.5 | 120.4±78.7 |
| | 0.001 | 0.01 | **134.7**±89.9 | **133.6**±89.9 | **134.9**±90.9 | **131.4**±86.2 | **136.9**±89.7 | **135.3**±85.1 | **138.0**±88.1 |
| Simple_Spread | 0 | 0 | -145.4±23.5 | -145.1±23.2 | -145.4±22.5 | -144.3±24.2 | -145.6±23.4 | -143.4±21.2 | -144.5±20.8 |
| | 0 | 0.01 | -139.1±24.8 | -139.1±24.8 | -138.7±24.6 | -138.7±25.2 | -138.8±24.4 | -139.7±23.9 | -140.0±24.0 |
| | 0.01 | 0 | -143.0±24.8 | -142.5±24.8 | -142.0±27.1 | -141.9±25.4 | -142.4±24.8 | -141.7±26.8 | -139.8±24.6 |
| | 0.01 | 0.01 | **-129.4**±20.1 | **-129.2**±19.9 | **-129.0**±20.2 | **-128.7**±19.2 | **-128.7**±20.3 | **-127.6**±19.2 | **-128.0**±20.6 |

# 6 Conclusion

Lossy communication remains a critical barrier hindering the practical deployment of MARL in real-world scenarios. To address this challenge, we propose a novel communication-constrained MARL framework that first establishes a unified prior over communication constraints via systematic modeling of lossy communication patterns, enabling agents to adapt strategies across diverse communication-constrained scenarios. Second, by distinguishing between lossy and lossless messages, we develop the Du-MIE to quantify the impact of messages on agent behavior, integrating this into the reward function to enhance the positive influence of reliable messages and mitigate the negative effects of corrupted messages. Finally, when integrated with the MADDPG algorithm, our approach demonstrates superior performance in both overall tasks and ablation studies across benchmark environments, validating its effectiveness in maintaining robust cooperative decision-making under varying communication constraints.

In the future, several directions warrant further investigation: (1) Algorithmic Scalability: Whether the framework can be extended to Value-based learning frameworks; (2) Adaptability to Dynamic Environments: Whether it can adaptively learn robust policies in highly dynamic communication-constrained environments?

## Acknowledgments and Disclosure of Funding

This work is supported in part by National Natural Science Foundation of China (62192783, 62276128, 62506157, 62276142), Jiangsu Natural Science Foundation (BK20243051), Jiangsu Science and Technology Major Project (BG2024031), the Fundamental Research Funds for the Central Universities (14380128, KG202514), Nanjing International/Hong Kong, Macao and Taiwan Science and Technology Cooperation Plan (202308031) and the Collaborative Innovation Center of Novel Software Technology and Industrialization.

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

# A  Appendices

## A.1  Details on Experiement

### A.1.1  Environmental Description

The experimental evaluation of this study was conducted in a widely used multi-agent particle environment(MPE). Specifically, the following three representative MPE scenarios were selected:

**Simple_Tag**: This scenario simulates the classic predator-prey pursuit game. $N$ predators with limited field of view radius ($N$ is set to 3 in this experiment, and the evaluation results of 6 and 9 agents are supplemented in the appendix) need to collaborate to capture a prey. The predators are controlled by independent reinforcement learning strategies, with larger body size, lower movement speed and smaller acceleration. The behavior strategy of the prey is predefined (randomly sampling 100 candidate positions, selecting the optimal movement direction within its perception range through the distance evaluation function). When any predator contacts the prey, all predators will receive a positive reward. In addition, 2 immovable obstacles are set in the environment to increase the complexity of the pursuit task.

**Simple_Spread**: This scenario requires $N$ agents (fixed to 3 in this experiment) to cover $N$ pre-set fixed landmarks through collaboration. Similar to Simple_Tag, each agent has a limited field of view. The reward for the agent is the negative of the sum of the minimum distances of all agents to their nearest uncovered landmarks. In addition, when there is a collision between agents, they will be penalized. Therefore, the agents must move quickly to cover their respective target landmarks, avoid collisions with each other, and ensure that all landmarks are uniquely covered.

**Simple_Reference**: This scenario is a unique collaborative navigation task, which includes 2 agents and 3 fixed landmarks of different colors. Unlike the previous two scenarios, each agent has global observation capabilities. The challenge is that the target landmark of each agent can only be known through the communication messages of another agent. The collective reward depends on the number of agents that reach the target landmark.

### A.1.2  Parameters Setting

In this experiment, an NVIDIA RTXA5000 24GB GPU was used. The actor network of each agent is a neural network with two hidden layers, each with 64 neurons, activated with ReLU, and the output layer with tanh activation function to output actions. All agents share a centralized critic network, whose hidden layer structure is similar to the actor network. In the JSD network, communication messages and actions are passed through a single-layer encoder with 32 neurons, respectively, and the mutual information lower bound is estimated using Jensen-Shannon divergence. In the CLUB network, the middle layer has 32 neurons, activated with ReLU, and the output layer uses tanh activation to model the conditional distribution of lossy messages and actions.

Adam optimizer is used for all networks. The learning rate of actor network, JSD network and CLUB network is $1 \times 10^{-4}$, the learning rate of critic network is $1 \times 10^{-3}$, the discount factor is set to 0.95, and the target network update rate is set to 0.01. The replay buffer size is $1 \times 10^5$, the message buffer size is $1 \times 10^3$, and the batch size is usually 1024 (in the Simple_Tag task, when the number of agents is 6 and 9, the batch size is adjusted to 512 to avoid the problem of CC-MADDPG training process exceeding GPU memory). The random seed is set to 1. The time step of each round is fixed to 25 steps. The total time step of training for all models is $4.0 \times 10^6$. When the total time steps exceed 1024, the model parameters are updated every 100 total time steps. For more effective action exploration, Ornstein-Uhlenbeck noise is added to the output actions of the actor network at the beginning of training with parameters $\theta = 0.15$ and $\sigma = 0.2$. At the beginning of training, the noise scale decays linearly with the number of training rounds.

During the evaluation, the trained model is loaded and run for 100 episodes in various test environments. The training process of each episode is fixed to 25 time steps. The average episode cumulative reward and standard deviation of each algorithm are mainly recorded and compared.

## A.2 More Experiemental Analysis

This study further explored the scalability of each algorithm when the number of agents increases. In the Simple_Tag scenario, theoretically, since there is no collision penalty between predators, increasing the number of predators should be able to directly increase the probability of capture and obtain higher cumulative rewards. However, the experimental results (see table 4) show that the increase in the number of agents also significantly increases the collaborative complexity of the task, which poses a challenge to all tested algorithms.

Table 4: Performance of Multi-Agent Algorithms under Varying Number of Agents and Communication Constraints

| Task Scenario | Testing Environment | Algorithms | | | | | |
| --- | --- | --- | --- | --- | --- | --- | --- |
| | | FC-MADDPG | Dropout-MADDPG | | | MADDPG | CC-MADDPG |
| | | | 0.2 | 0.5 | 0.8 | | |
| Simple_Tag (3 agents) | Unrestricted | 75.9±65.3 | 70.3±65.9 | 65.9±62.0 | 72.1±66.1 | 5.2±12.8 | **134.7**±89.9 |
| | Light MBC (3) | 72.9±65.3 | 70.9±65.9 | 65.9±62.1 | 72.5±66.1 | 5.2±12.8 | **133.6**±89.9 |
| | Medium MBC (6) | 67.2±53.7 | 71.0±65.7 | 67.1±62.3 | 72.1±66.1 | 5.5±15.6 | **134.9**±90.9 |
| | Heavy MBC (8) | 54.4±43.8 | 69.8±66.5 | 67.9±63.2 | 72.7±66.0 | 6.0±14.2 | **131.4**±86.2 |
| | Light DBC (5) | 19.5±38.2 | 69.0±65.8 | 67.3±62.4 | 70.8±64.2 | 23.5±41.5 | **136.9**±89.7 |
| | Medium DBC (3) | 10.9±26.5 | 70.1±65.8 | 67.8±62.4 | 70.7±63.8 | 44.8±58.4 | **135.3**±85.1 |
| | Heavy DBC (1) | 1.5±5.2 | 68.7±64.3 | 66.3±60.5 | 71.4±65.7 | 71.2±70.9 | **138.0**±88.1 |
| Simple_Tag (6 agents) | Unrestricted | 138.5±88.0 | 84.2±70.3 | 78.5±68.1 | 73.1±61.6 | 6.2±12.8 | **131.8**±89.9 |
| | Light MBC (3) | 123.9±77.0 | 83.5±70.3 | 78.1±67.3 | 72.9±61.3 | 6.2±13.1 | **131.9**±90.3 |
| | Medium MBC (6) | 78.5±63.1 | 82.1±70.9 | 79.6±67.9 | 72.0±64.3 | 6.4±12.8 | **133.2**±84.3 |
| | Heavy MBC (8) | 69.8±59.5 | 81.3±69.0 | 78.6±65.4 | 75.2±61.8 | 7.0±15.3 | **131.8**±85.8 |
| | Light DBC (5) | 14.7±31.0 | 83.0±71.0 | 80.1±68.6 | 75.7±62.5 | 6.6±11.8 | **135.3**±89.4 |
| | Medium DBC (3) | 5.4±13.4 | 75.5±66.3 | 79.3±64.4 | 77.2±63.5 | 8.0±14.5 | **127.4**±89.1 |
| | Heavy DBC (1) | 3.8±10.0 | 79.1±65.3 | 72.2±69.3 | 76.5±63.7 | 11.1±20.9 | **122.1**±77.8 |
| Simple_Tag (9 agents) | Unrestricted | 83.4±72.5 | 77.7±71.5 | 68.0±61.0 | 19.8±27.7 | 7.2±15.4 | **78.2**±78.1 |
| | Light MBC (3) | 77.9±69.5 | 77.8±71.9 | 68.1±61.1 | 19.5±30.4 | 7.3±15.4 | **77.7**±72.6 |
| | Medium MBC (6) | 50.3±64.5 | 77.0±71.7 | 66.3±60.9 | 19.0±25.5 | 8.3±12.7 | **78.9**±72.5 |
| | Heavy MBC (8) | 38.9±35.3 | 77.4±72.5 | 67.3±61.9 | 16.5±28.0 | 8.1±17.0 | **77.6**±73.4 |
| | Light DBC (5) | 9.7±18.6 | 42.9±53.2 | 68.9±64.6 | 19.9±36.7 | 13.9±22.5 | **82.7**±74.4 |
| | Medium DBC (3) | 8.0±16.5 | 27.1±41.6 | 56.3±60.5 | 20.4±33.7 | 18.1±32.5 | **83.7**±71.0 |
| | Heavy DBC (1) | 5.8±10.7 | 21.0±33.4 | 49.0±52.7 | 21.1±34.9 | 17.5±30.8 | **83.2**±70.6 |

Specifically, although FC-MADDPG achieved an average reward of up to 138.5 in an ideal environment with 6 agents, its performance dropped significantly after the introduction of constraints or when the number of agents increased, once again proving its dependence on ideal communication and its limitations under complex coordination. For the Dropout-MADDPG series of algorithms, its robustness advantage weakens when the number of agents increases, and it is difficult to cope with environments with high collaborative complexity and strong communication constraints. For example, in the scenario of 9 agents, the average rewards of dropout-0.2 in distance-based constraint are reduced to 42.9, 27.1 and 21.0 respectively.

In contrast, CC-MADDPG exhibits stronger robustness and scalability. In the scenarios of 6 and 9 agents, its performance in an ideal environment is close to that of FC-MADDPG, and it always maintains a leading performance level after the introduction of communication constraints. Secondly, when the number of agents increases, its robustness advantage over the simple message dropout method is more obvious in a constrained environment. For example, in the scenario of 9 agents, under the heavy distance-based constraint, CC-MADDPG can achieve an average reward of 83.2, while dropout-0.5 is only 49.0. This shows that CC-MADDPG is more robust and effective than the simple message dropout method when dealing with more complex communication-constrained problems.

