# OpenReview forum: "Multi-Agent Reinforcement Learning with Communication-Constrained Priors"
_NeurIPS.cc/2025/Conference — NeurIPS 2025 poster_

### Official Review · Reviewer_c2rM · 2025-07-02

**Clarity:** 3
**Significance:** 3
**Originality:** 3
**Rating:** 4
**Confidence:** 4

**Summary:**

The paper proposes a communication-constrained multi-agent reinforcement learning (MARL) framework to address the challenges of lossy communication in real-world scenarios. It introduces a generalized model to characterize communication conditions and uses this model as a learning prior to distinguish between lossy and lossless messages. The framework employs a dual mutual information estimator (Du-MIE) to quantify the impact of messages on agent behaviors, enhancing the positive effects of reliable messages and mitigating the negative effects of unreliable ones. The proposed approach is validated through experiments in various communication-constrained environments, demonstrating superior performance and robustness compared to existing methods.

**Questions:**

Please refer to the section "Weaknesses" for detailed information.

**Ethical Concerns:**

["NO or VERY MINOR ethics concerns only"]

**Final Justification:**

**Final Decision**: I decide to raise my score to 4.
**Justification**: The paper proposes a communication-constrained multi-agent reinforcement learning (MARL) framework to address the challenges of lossy communication in real-world scenarios. Overall, the writing is clear, the motivation is relatively well-defined, and the experiments are reasonably comprehensive. During the rebuttal phase, the issues that have been addressed through active interaction with the authors include the discussion on motivation and the addition of experiments in large-scale scenarios. The issue that remains unresolved is that the experimental scenarios are somewhat simple and need validation in more diverse and complex environments. Taking into account the authors' responses and other reviewers' comments, I decide to raise my score to 4.

**Limitations:**

Yes.

**Paper Formatting Concerns:**

No major formatting issues in this paper.

**Quality:**

2

**Strengths And Weaknesses:**

# Strengths
1. The paper is reasonably well-written, with a clearly defined research problem and a clearly stated motivation. The overall structure is logical and the paper is generally easy to follow.
2. The paper exhibits a certain level of originality.
3. The paper demonstrates a certain degree of originality and is also highly readable, particularly Section 4.

# Weaknesses
1. Regarding the motivation, the main issue the paper aims to address is the problem of communication loss. Therefore, mentioning the issue of limited communication bandwidth (line 23-30) may be off-topic and could dilute the focus of the study. It is recommended to remove this part.
2. The Introduction mentions that existing methods suffer from poor scalability. To better support this claim, it is suggested to include evaluations on larger-scale scenarios. Additional relevant experiments could be included to strengthen the paper.
3. Another suggestion is to include a visualization section in Section 5 to further demonstrate the effectiveness of the proposed method, which would make the work more comprehensive.
4. There are also minor issues, such as missing symbols in Equation (2)(6). The authors are advised to conduct a thorough review for such details.

If the authors can address this, I would consider upgrading my score.

---

> ### Author Rebuttal · Authors · 2025-07-31
>
> We sincerely thank the reviewer for efforts and the useful suggestions. We have prioritized and organized the reviewer’s comments, and our responses are as follows:
>
> S1. It is suggested to include evaluations on larger-scale scenarios.
>
> R1. Thank you for the suggestion. We have supplemented Appendix A.2 with results for settings with 6 and 9 agents. In addition, we also add an experiment with 24 agents, presented below. The results indicate that as the number of agents increases, our method maintains its advantage.
>
> | simple_tag(24 agents) | fc-maddpg | maddpg-dropout-0.2 | maddpg-dropout-0.5 | maddpg-dropout-0.8 | maddpg | cc-maddpg |
> | --- | --- | --- | --- | --- | --- | --- |
> | unrestricted | 77.5±64.4 | 13.4±18.5 | 10.6±15.9 | 13.1±16.3 | 8.1±13.2 | 82.9±69.9|
> | Light MBC(3) | 77.3±64.5 | 13.5±17.2 | 10.6±15.9 | 13.1±16.2 | 8.2±13.2 | 82.6±69.2|
> | Medium MBC(6) | 67.3±51.0 | 14.1±19.5 | 9.5±14.4 | 13.2±16.4 | 7.9±13.1 | 83.6±69.9|
> | Heavy MBC(8) | 53.5±46.4 | 13.8±18.7 | 10.9±18.0 | 13.0±16.5 | 8.0±13.4	| 85.0±71.5|
> | Light DBC(5) | 14.5±23.4 | 13.5±17.2 | 11.9±19.4 | 12.9±16.0 | 9.7±16.2 | 87.5±71.9|
> | Medium DBC(3) | 12.9±23.8 | 14.7±22.2 | 12.5±18.6 | 13.4±17.0 | 9.3±15.5 | 88.1±75.4|
> | Heavy DBC(1) | 10.5±19.2 | 13.6±20.3 | 12.2±18.1 | 14.4±20.6 | 9.0±14.9	| 92.0±75.7|
>
> S2. Another suggestion is to include a visualization section in Section 5.
>
> R2. Thank you for the suggestion. Our goal is to address the performance degradation that arises when deploying models in lossy communication scenarios, thus robustness is the core metric we intend to highlight. We believe that a tabular format can more clearly convey the robustness achieved under various lossy communication scenarios. Nevertheless, we will adopt your suggestion. We have generated training curves and a bar chart comparing the final rewards based on the log data. These figures will be presented in the revised appendix.
>
> S3. It is recommended to remove the related work mentioning the issue of limited communication bandwidth.
>
> R3. Thank you for the suggestion. We will remove in the revised version.
>
> S4. The authors are advised to conduct a thorough review for minor issues, such as missing symbols in Equation (2)(6).
>
> R4. Thank you for the suggestion. We will carefully ensure equation consistency in the revised version.

---

> > ### Comment · Reviewer_c2rM · 2025-08-06
> >
> > Thank you for your response! Your answers have addressed my concerns, and I will increase my score. Additionally, I also interested in the reviewer fm9D's comment: MPE is indeed relatively simple; adding more challenging environments, such as SMAC, would be a reasonable and valuable extension.

---

> > > ### Author Response · Authors · 2025-08-06
> > >
> > > We sincerely appreciate the reviewer's efforts and the decision to increase the paper's score. We are also grateful for the valuable suggestion regarding the extensibility of the paper's experiments. We will further supplement the SMAC results in the final submitted version.
> > >
> > > Best regards,
> > > 27091 Authors

---

### Official Review · Reviewer_UbAw · 2025-07-02

**Clarity:** 4
**Significance:** 3
**Originality:** 3
**Rating:** 4
**Confidence:** 4

**Summary:**

This paper investigates the problem of lossy communication in communicative multi-agent reinforcement learning. To overcome the challenges, a natural goal is to maximize the utilization of lossless messages while minimizing the adverse effects of lossy messages. This paper proposes a novel framework that incorporates communication-constrained priors to model the lossy communication and uses a dual mutual information estimator (Du-MIE) to optimize the impact of messages on agent behaviors. The learned policies can efficiently communicate and collaborative act in multi-agent scenarios with lossy communication. The experimental results demonstrate the effectiveness of the proposed approach in maintaining robust cooperative decision-making under varying communication constraints in MBC and DBC.

**Questions:**

Q1: Can the proposed method still work well in cooperative scenarios where the agent can defect for self-interest?

Q2: It seems one key related paper on limited bandwidth with information bottleneck is missing: "Wang et al. 2020. Learning Efficient Multi-agent Communication: An Information Bottleneck Approach."

**Ethical Concerns:**

["NO or VERY MINOR ethics concerns only"]

**Limitations:**

See above comments

**Paper Formatting Concerns:**

None.

**Quality:**

3

**Strengths And Weaknesses:**

**Strength:**
1. This paper investigates an interesting lossy communication problem in MARL and introduces a novel communication-constrained MARL framework that effectively addresses the challenges of lossy communication in multi-agent systems.
2. The proposed method can be applied to many MARL methods, showing its robustness and adaptability.
3. The authors provide extensive experimental results on MBC and DBC.
4. This paper is well-written and clearly organized.

**Weakness:**
1. It is not clear the scalability capacity of the proposed method in multi-agent scenarios where the number of agents is huge.

---

> ### Author Rebuttal · Authors · 2025-07-31
>
> We sincerely thank the reviewer for the time and efforts. Below please find the responses to some specific comments.
>
> Q1. Can the proposed method still work well in cooperative scenarios where the agent can defect for self-interest?
>
> A1. Our proposed method does not specifically address this issue. Our approach targets the alleviate the degradation of MARL performance under lossy communication, whereas cooperative scenarios in which agents may defect for self-interest—often termed social dilemmas—require dedicated methods (e.g., incorporating social preferences), which is out of our scope. We are happy to investigate the influence of lossy communication in such scenarios as future work
>
> Q2. It seems one key related paper on limited bandwidth with information bottleneck is missing: "Wang et al. 2020. Learning Efficient Multi-agent Communication: An Information Bottleneck Approach."
>
> A2. Thank you very much for the reminder. We will add the relevant work in the revised version.
>
> W1. It is not clear the scalability capacity of the proposed method in multi-agent scenarios where the number of agents is huge.
>
> R1. Thank you for pointing out this. We have supplemented Appendix A.2 with results for settings with 6 and 9 agents. In addition, for verity the scalability capacity, we also add an experiment with 24 agents, presented below. The results indicate that as the number of agents increases, our method maintains its advantage.
>
> | simple_tag(24 agents) | fc-maddpg | maddpg-dropout-0.2 | maddpg-dropout-0.5 | maddpg-dropout-0.8 | maddpg | cc-maddpg |
> | --- | --- | --- | --- | --- | --- | --- |
> | unrestricted | 77.5±64.4 | 13.4±18.5 | 10.6±15.9 | 13.1±16.3 | 8.1±13.2 | 82.9±69.9|
> | Light MBC(3) | 77.3±64.5 | 13.5±17.2 | 10.6±15.9 | 13.1±16.2 | 8.2±13.2 | 82.6±69.2|
> | Medium MBC(6) | 67.3±51.0 | 14.1±19.5 | 9.5±14.4 | 13.2±16.4 | 7.9±13.1 | 83.6±69.9|
> | Heavy MBC(8) | 53.5±46.4 | 13.8±18.7 | 10.9±18.0 | 13.0±16.5 | 8.0±13.4	| 85.0±71.5|
> | Light DBC(5) | 14.5±23.4 | 13.5±17.2 | 11.9±19.4 | 12.9±16.0 | 9.7±16.2 | 87.5±71.9|
> | Medium DBC(3) | 12.9±23.8 | 14.7±22.2 | 12.5±18.6 | 13.4±17.0 | 9.3±15.5 | 88.1±75.4|
> | Heavy DBC(1) | 10.5±19.2 | 13.6±20.3 | 12.2±18.1 | 14.4±20.6 | 9.0±14.9	| 92.0±75.7|

---

> ### Comment · Reviewer_UbAw · 2025-08-07
> **Response to authors**
>
> Thanks for the responses.

---

### Official Review · Reviewer_fm9D · 2025-07-03

**Clarity:** 3
**Significance:** 3
**Originality:** 2
**Rating:** 4
**Confidence:** 3

**Summary:**

This paper proposes a communication-constrained MARL framework that introduces: (1) a generalized binary link model to distinguish lossy/lossless messages across diverse scenarios; (2) a dual mutual information estimator (Du-MIE) using JSD to amplify useful messages and CLUB to suppress noisy ones; and (3) a reward-shaping mechanism integrating MI estimates into global rewards. Validated as CC-MADDPG in particle environments with Markov/distance-based constraints, it achieves significant performance gains by maintaining robust cooperation under dynamic communication degradation.

**Questions:**

1.See the weaknesses.

2.During practical implementation, how is the agent-pair-specific state component sij
distilled from the global state s? Specifically, what key information dimensions does sij encompass?

3.Would significant imbalance in the ratio between lossless and lossy messages lead to substantial performance degradation?

**Ethical Concerns:**

["NO or VERY MINOR ethics concerns only"]

**Final Justification:**

The author's response has addressed most of my concerns. Hence, I improve my rating to 4.

**Limitations:**

yes

**Quality:**

2

**Strengths And Weaknesses:**

Strengths:

1.The paper is well written and easy to read, with the method/motivation/story communicated clearly to the reader.

2.The paper employs a Dual Mutual Information Estimator (Du-MIE) to optimize the multi-agent communication mechanism, a framework that demonstrates both theoretical soundness and practical feasibility.

3.This method framework represents a significant improvement over the baseline.

Weaknesses:

1.This study demonstrates a fundamental divergence from PMIC in core research focus: while PMIC primarily concentrates on optimizing multi-agent collaborative behaviors, this work specifically addresses the critical challenge of message transmission reliability in communication-constrained scenarios. Although differing in research objectives, both approaches employ the Dual Mutual Information Estimator (Du-MIE) as the key technical component. It should be noted that the methodology substantially inherits PMIC's foundational framework, and this design choice has, to some extent, constrained the innovative demonstration of this research.

2.While the methodological framework of this paper demonstrates sound logic, the experimental design exhibits several limitations:

a)Inadequate baseline comparison: The significant improvement over MADDPG alone is insufficient to substantiate performance superiority. Comparative experiments with other mutual information-based communication optimization methods (e.g., NDQ, MAIC) should be included.

b)Limited test scenarios: The validation using only three cooperative scenarios from MPE lacks adversarial/competitive environment testing, which constrains the generalizability of conclusions.

c)Oversimplified test conditions: The observed "relatively insignificant performance variations across different network fluctuation levels" may stem from undemanding test environments. More challenging experimental settings should be employed to verify this finding.

---

> ### Author Rebuttal · Authors · 2025-07-31
>
> We sincerely thank the reviewer for the time and efforts. Below please find the responses to some specific comments.
>
> Q1. During practical implementation, how is the agent-pair-specific state component sij distilled from the global state s? Specifically, what key information dimensions does sij encompass?
>
> A1. Thank you for your concern. In MPE, the environment state is formed by concatenating the local observations of all agents. Simply splitting this vector yields the entries relevant to agents i and j, notably their respective spatial positions and velocities.
>
> Q2. Would significant imbalance in the ratio between lossless and lossy messages lead to substantial performance degradation?
>
> A2. Thank you for your concern. It indeed leads to degraded performance, as demonstrated by our experiments. We configured two lossy communication scenarios, MBC and DBC, correspondingly set two parameters—loss probability and distance threshold—that govern the ratio between lossless and lossy messages. We further stratified the severity of communication loss into three levels: high, medium, and low. As shown in Table 1, FC-MADDPG with full communication performs worse when the proportion of lossy messages exceeds that of lossless messages, whereas the no-communication MADDPG deteriorates when the proportion of lossy messages is lower than that of lossless messages due to interference from redundant messages.
> We attribute the performance degraded mainly to the mismatch between the message distributions at training and testing. Specifically, FC-MADDPG is trained under full communication yet evaluated under lossy communication, while MADDPG is trained without communication but tested in the same lossy communication. We will add the analysis in the revised version.
>
>
> W1. Inadequate baseline comparison.
>
> R1. Thank you for the comment. We have conducted new experiments to compare with the baselines the reviewer suggested. However, we want to clarify that NDQ is incompatible since it addresses limited communication bandwidth and cannot be applied to lossy communication settings. MAIC is designed for discrete action spaces, accordingly, we discretized the MPE setting and have
>
> | Scenarios| Unrestricted	| Light MBC(3) | Medium MBC(6) | Heavy MBC(8) | Light DBC(5) | Medium DBC(3) | Heavy DBC(1) |
> | --- | --- | --- | --- | --- | --- | --- | --- |
> | simple\_tag | 2.7±9.1 | 2.4±6.5 | 2.3±7.8 | 1.7±6.2 | 1.6±8.6 | 1.8±6.0 | 1.5±4.9 |
> | simple\_reference | 0.2±1.4	 | 0.3±1.7 | 0.2±1.4 | 0.2±1.3 | 0.4±2.0 | 0.3±1.4 | 0.1±1.0|
> | simple\_spread | -298.0±75.1 | -293.3±63.8 | -295.9±78.4 | -283.7±51.8 | -301.2±65.5 | -282.4±63.8 | -289.5±63.1 |
> | simple\_adversary |-29.5±26.4 | -33.9±26.7 | -35.4±29.5 | -33.1±28.7 | -34.9±31.0 | -29.3±25.4 | -35.0±29.3 |
>
> Experimental results indicate that MAIC performs worse than our proposed method under these lossy-communication scenarios.
>
> W2. Limited test scenarios.
>
> R2. Thank you for the comment. We add an experiment on a cooperative–competitive scenario (simple_adversary), and the results show that our proposed method likewise exhibits a performance advantage.
>
> | simple_adversary | fc-maddpg | maddpg-dropout-0.2 | maddpg-dropout-0.5 | maddpg-dropout-0.8 | maddpg | cc-maddpg |
> | --- | --- | --- | --- | --- | --- | --- |
> | Unrestricted	| -6.7±5.1 | -6.8±5.0 | -6.6±4.8 | -6.7±4.8 | -75.5±48.5 | -5.8±5.0 |
> | Light MBC(3) | -6.7±5.1 | -6.8±5.0 | -6.6±4.8 | -6.7±4.8 | -75.5±48.5 | -5.9±5.0 |
> | Medium MBC(6) | -7.1±5.0 | -7.2±5.1 | -6.6±4.9 | -6.7±4.8	| -62.9±41.6 | -6.1±5.0 |
> | Heavy MBC(8) | -7.5±5.0 | -6.8±5.0 | -6.8±4.9 | -6.7±4.7 | -51.5±34.5 | -6.1±5.0 |
> | Light DBC(5) | -26.6±15.1 | -10.8±13.3 | -6.3±5.0 | -6.7±4.7 | -14.0±8.0 | -6.4±5.8 |
> | Medium DBC(3) | -38.6±21.3 | -22.4±19.0 | -6.9±5.2 | -6.7±4.7 | -8.8±5.8 | -6.7±6.2 |
> | Heavy DBC(1) | -45.3±21.8 | -27.3±19.5 | -7.7±5.3 | -6.8±4.7 | -7.8±5.5 | -5.8±5.4 |
>
>
> W3 (original Q2). Oversimplified test conditions
>
> R3. Thank you for pointing out this. We would like to clarify this assumption in two points. First, this test conditions pose a significant challenge to conventional communication-based MARL. As shown in Table 1, their performance degrades markedly as the proportion of lost messages increases. Moreover, the test conditions (binary-like) is also partially employed in the validation of the MAIC method.
> Second, it aligns well with real-world scenarios that the communication could be broken occasionally, e.g., For example, during network transmission, data are segmented into individual packets. Packet loss occurs when one or more packets are lost in transit and fail to reach the destination, yielding an inherently binary outcome—either the packet arrives or it does not. Likewise, in underwater communication among UUVs, once the distance exceeds a certain threshold, message transmission is naturally dropped, producing the same binary-like outcome.
>
> W4. It should be noted that the methodology substantially inherits PMIC's foundational framework, and this design choice has, to some extent, constrained the innovative demonstration of this research.
>
> R4. Thank you for the comment. We believe that the novelty of our method is by no means limited to this single aspect. We are the first to formalize lossy communication and to propose an innovative communication-constrained prior model. Leveraging a Dual-MIE scheme, we simultaneously (1) mitigate the adverse effects of lossy messages and (2) amplify the beneficial influence of lossless messages. Both the modeling framework and the solution procedure make a substantial contribution to MARL with lossy communication research and facilitate the rapid transfer of MARL techniques to real-world applications.
>
> Moreover, our work is not a mere duplication of PMIC. Two clarifications are in order:
>
> (1) Problem scope. We address robust policy learning under lossy communication, whereas PMIC focuses on solving overgeneralization issues in MASs.
>
> (2) Methodological framework. Although both approaches employ JSD and CLUB as MIE, their roles differ markedly: we measure the MI between messages and individual behaviors to control the influence of communication on policy training, while PMIC uses it to distinguish the optimal trajectories and sub-optimal ones. Consequently, our method does not require the additional dual buffer used in PMIC.

---

> > ### Comment · Reviewer_fm9D · 2025-08-07
> >
> > Thank you for response. It has addressed most of my concerns, and I will take these clarifications into account when finalizing my rating.

---

> > > ### Author Response · Authors · 2025-08-07
> > >
> > > We are truly grateful that our response has addressed most of your concerns. We would like to once again express our gratitude to the reviewer for the time and efforts dedicated to our paper, as well as for the valuable suggestions provided on it!
> > >
> > > Best regard,
> > >
> > > All authors in submission27091

---

### Official Review · Reviewer_ZcE1 · 2025-07-03

**Clarity:** 2
**Significance:** 3
**Originality:** 3
**Rating:** 5
**Confidence:** 5

**Summary:**

This paper proposes a novel communication-constrained multi-agent reinforcement learning (MARL) framework designed to address lossy and unreliable communication in real-world environments. The authors introduce a unified communication prior model to distinguish between lossless and lossy messages across diverse scenarios and quantify their impact on decision-making via a dual mutual information estimator (Du-MIE). By shaping the reward with mutual information signals, the framework enables robust policy learning under varying communication conditions. Experimental results on MPE benchmarks demonstrate the method's effectiveness and generalization compared to baseline MARL algorithms.

**Questions:**

1. **Is the binary communication prior overly simplified?**
   The prior $\iota_{ij} \in \{0,1\}$ models message reception as either fully successful or completely failed. However, in many practical systems, messages may be partially received, noisy, or distorted. Can the current framework handle such intermediate or stochastic communication states?

2. **Why is MADDPG used as the base algorithm?**
   Would the proposed Du-MIE framework still demonstrate consistent advantages if built on stronger backbones like MAPPO? An ablation using a more competitive base algorithm would strengthen the claim of generality.

3. **What is the training overhead of Du-MIE?**
   Since both the JSD and CLUB modules are trained jointly with the policy, how much additional computational cost do they introduce? A comparison of training efficiency with and without Du-MIE would be helpful for understanding practical applicability.

**Ethical Concerns:**

["NO or VERY MINOR ethics concerns only"]

**Final Justification:**

The authors have addressed most of my initial concerns in the rebuttal:

- Justified the binary communication prior with real-world examples.
- Explained the choice of MADDPG and committed to adding MAPPO results.
- Provided training overhead details and acknowledged notation inconsistencies.
- Agreed to include training curves and expand related work.

Some concerns remain (e.g., stronger baseline comparisons), but the paper is technically solid and the authors show willingness to improve.

**Limitations:**

The authors have adequately addressed the limitations.

**Paper Formatting Concerns:**

No major formatting issues found.

**Quality:**

3

**Strengths And Weaknesses:**

**Strengths**
1. The dual mutual information estimator (Du-MIE) is a novel design that separately optimizes for lossless and lossy messages, enabling more robust policy learning.
2. Extensive experiments across different constrained environments show consistent improvements over competitive baselines.
3. The paper is well structured and clearly presented, with detailed ablations and intuitive motivation.

**Weaknesses**

Despite its novel integration of communication modeling and message utility estimation, the paper appears somewhat rushed in presentation, with several notable issues:
1. **Inconsistent Mathematical Notation**: The notation lacks rigor and clarity. For instance, $r = R(s, a)$ should use bold $\boldsymbol{a}$ to indicate a joint action, and the policy set $\pi = \{ \pi_1, ..., \pi_N \}$ should use bold $\boldsymbol{\pi}$ for consistency. Such issues appear repeatedly and detract from the formal precision.
2. **Limited Related Work and Baseline Coverage**:  The related work section is brief and omits prior efforts on communication under noise or loss. Including these works and comparing against stronger baselines would enhance fairness and context.
3. **Lack of Experimental Detail and Visualization**:  The paper does not specify the number of seeds or clarify whether $ \text{mean} \pm \text{std} $ refers to standard deviation. The absence of training curves also hinders evaluation of convergence and stability.

---

> ### Author Rebuttal · Authors · 2025-07-31
>
> We sincerely thank the reviewer for the time and efforts. Below please find the responses to some specific comments.
>
> Q1. Is the binary communication prior overly simplified?
>
> A1. Thank you for pointing out this. We would like to clarify this assumption in two points. First, it aligns well with real-world scenarios that the communication could be broken occasionally, e.g., For example, during network transmission, data are segmented into individual packets. Packet loss occurs when one or more packets are lost in transit and fail to reach the destination, yielding an inherently binary outcome—either the packet arrives or it does not. Likewise, in underwater communication among UUVs, once the distance exceeds a certain threshold, message transmission is naturally dropped, producing the same binary-like outcome.
> Second, this binary-like abstraction of realistic communication scenarios poses a significant challenge to conventional communication-based MARL. As shown in Table 1, their performance degrades markedly as the proportion of lost messages increases. Moreover, the binary-like abstraction is also partially employed in the validation of the MAIC method.
>
> Q2. Why is MADDPG used as the base algorithm?
>
> A2. Thank you for your concern. We choose MADDPG as the base algorithm for two reasons. First, it embodies both the AC architecture and CTED framework, making it representative of many MARL methods. Second, MADDPG integrates naturally with communication mechanisms without requiring additional modifications, enabling ablation studies to cleanly verify the effectiveness of each module in our approach. By contrast, in standard MAPPO the actor network employs an RNN; when combined with a communication mechanism, this typically could necessitate designing new neural network components, making it difficult to disentangle whether performance changes arise from the neural network architecture or from our algorithmic modules. While our method is orthogonal to existing MARL approaches in principle. We will include more results on the backbone of other MARL algorithms, such as MAPPO in our revised version.
>
> Q3. What is the training overhead of Du-MIE?
>
> A3. We report the training time for each algorithm over 10 runs, with 4 million steps per run. All runs were conducted on the same server equipped with an NVIDIA A5000 GPU.
>
> | Scenarios | fc-maddpg | maddpg-dropout-0.2 | maddpg-dropout-0.5 | maddpg-dropout-0.8 | maddpg | cc-maddpg |
> | --- | --- | --- | --- | --- | --- | --- |
> | simple_tag | 5.71hr | 5.42hr | 6.61hr | 6.56hr | 8.03hr | 14.17hr |
> | simple_reference | 3.03hr | 3.24hr | 4.87hr | 3.03hr | 3.06hr | 5.95hr |
> | simple_spread | 4.13hr | 4.35hr | 8.49hr | 4.97hr | 8.21hr | 13.31hr |
>
> It is noted that the Du-MIE module doesn’t induce heavy workload of training resources, considering the advantages contributed to the final performance.
>
> W1. Limited Related Work and Baseline Coverage.
>
> R1. Thank you for your comment. We have investigated the degradation of MARL performance under lossy communication. However, there are quite few works in this domain. Approaches that assume noisy (rather than lossy) communication differ from our problem formulation: Approaches with noisy communication assume that noise can be modeled and noisy messages can be repaired, while we assume that messages are directly lost and messages are unusable. Thus, approaches with noisy communication not suitable as baselines.
>
> W2. Lack of Experimental Detail and Visualization.
>
> R2. Thank you for your comment. For training our algorithm, we used random seed 1(no average over at least 5 seeds), which yielded the best performance. During evaluation, we did not fix the random seed.
> Moreover, our goal is to address the performance degradation that arises when deploying models in lossy communication scenarios, and robustness is the core metric we intend to highlight.
> We believe that, instead of presenting numerous training curves for different tasks, a tabular format can more clearly convey the robustness achieved under various lossy communication scenarios.
> Nevertheless, we will adopt your suggestion. We have compiled the training curves and will present them in the appendix.
>
> W3. Inconsistent Mathematical Notation.
>
> R3. Thank you for the comment. We will carefully ensure notation consistency in the final version.

---

> > ### Comment · Reviewer_ZcE1 · 2025-08-04
> >
> > Thank you for the detailed rebuttal. Your response addressed my concerns and clarified the motivation and design choices effectively. I appreciate the additional explanations and experimental details. I will take this into consideration when updating my ranking.

---

> > > ### Author Response · Authors · 2025-08-07
> > >
> > > We are truly grateful that our response has addressed most of your concerns. Meanwhile, we would like to once again express our gratitude to the reviewer for the time and efforts dedicated to our paper!
> > >
> > > Best regards,
> > >
> > > All authors

---

### Comment · Area_Chair_JPZ7 · 2025-08-05

Dear Reviewers

Thanks for participating in discussion. The author-reviewer discussion period is extended to Aug. 8, 11:59PM AoE.

If you have not yet, I kindly ask you to

- Read the author rebuttal and other reviews.
- Post your response to the author rebuttal as soon as possible.
- Engage in discussion with the authors and other reviewers where necessary.

It is not permitted by the conference policy that a reviewer submits mandatory acknowledgement without a single reply sentence to author rebuttal.

If you have already engaged in author-reviewer discussion, I sincerely thank you. If not, I kindly ask you to engage in discussion.

Thank you very much for your help.

Best regard,

AC

---

### Comment · Area_Chair_JPZ7 · 2025-08-08

Dear Reviewers

Thank you again for participating in reveiwer-author discussion.

The reviewer-author discussion is coming to an end soon. If you did not submit mandatory acknowledgement yet, please submit the ack, final score and justification by the end of the reviewer-author discussion period.

Thanks.

Area Chair

---

### Decision · Program_Chairs · 2025-09-17

**Decision:**

Accept (poster)

**Comment:**

The paper proposes a systematic training method to cope with communication loss in multi-agent reinforcement learning with communication. With the availability assumption of ideal communication at the training phase, the authors used the communication loss prediction model based on communication link modeling. Then, the authors identified correctly received messages and lost messages and generated a shaped reward for training, boosting the impact of correct message and suppressing the impact of lost messages, based on bounds on mutual information. The idea flow is natural and systematic, and the proper use of lower and upper bounds of mutual information is interesting. The proposed method shows better performance than previous message-dropout. So, AC recommends the acceptance of the paper.